# Use of Dexpanthenol for Atopic Dermatitis—Benefits and Recommendations Based on Current Evidence

**DOI:** 10.3390/jcm11143943

**Published:** 2022-07-06

**Authors:** Yoon Sun Cho, Hye One Kim, Seung Man Woo, Dong Hun Lee

**Affiliations:** 1Bayer Korea Consumer Health, Seoul 07335, Korea; yoonsun.cho@bayer.com; 2Department of Dermatology, Hallym University Kangnam Sacred Heart Hospital, Seoul 07441, Korea; hyeonekim@gmail.com; 3Ewha Skin & Laser Clinic, Seoul 06912, Korea; drw00@naver.com; 4Department of Dermatology, Seoul National University Hospital, Seoul National University College of Medicine, Seoul 03080, Korea

**Keywords:** atopic dermatitis, dexpanthenol, topical corticosteroid, emollient

## Abstract

Background: Atopic dermatitis (AD) is an inflammatory skin disease of multiple phenotypes and endotypes, and is highly prevalent in children. Many people of all ages, including active adolescents, pregnant women, and the elderly, suffer from AD, experiencing chronicity, flares, and unexpected relapse. Dexpanthenol has multiple pharmacological effects and has been employed to treat various skin disorders such as AD. We aimed to summarize the up-to-date evidence relating to dexpanthenol and to provide a consensus on how to use dexpanthenol effectively for the treatment of AD. Methods: The evidence to date on the application and efficacy of dexpanthenol in AD was reviewed. The literature search focused on dexpanthenol use and the improvement of skin barrier function, the prevention of acute flares, and its topical corticosteroid (TCS) sparing effects. Evidence and recommendations for special groups such as pregnant women, and the effects of dexpanthenol and emollient plus in maintenance therapy, were also summarized. Results: Dexpanthenol is effective and well-tolerated for the treatment of AD. Dexpanthenol improves skin barrier function, reduces acute and frequent flares, has a significant TCS sparing effect, and enhances wound healing for skin lesions. Conclusion: This review article provides helpful advice for clinicians and patients on the proper maintenance treatment of AD. Dexpanthenol, as an active ingredient in ointments or emollients, is suitable for the treatment and maintenance of AD. This paper will guide dermatologists and clinicians to consider dexpanthenol as a treatment option for mild to moderate AD.

## 1. Introduction

Atopic dermatitis (AD) is a chronic inflammatory skin disorder characterized by pruritus and recurrent inflammation [1]. Patients with AD commonly suffer from many symptoms and signs, including pruritis, pain, erythema, excoriation, and sleep disturbances [2,3,4]. Difficulties during outdoor activities and periorbital hyperpigmentation are common complaints of active adolescents and women, respectively [5,6,7,8]. The treatment of AD might be particularly challenging for certain patients, including those with severe AD [9,10,11,12,13], frequent relapses, extensive area of involvement [14,15], and steroid-phobia, those without experience of full recovery, thereby making rapport difficult [16,17], and the elderly with weak, thin skin.

Most guidelines recommend the use of emollients in conjunction with topical corticosteroids (TCS) for the initial treatment of this intractable disease. Moisturizer is used synonymously with emollient and refers to as a product that moisturizes and smooths the skin, whereas humectant increases or maintains hydration of the skin [18]. Dexpanthenol is a stable alcohol analog of pantothenic acid (vitamin B5) with moisturizing and wound healing efficacy. Increasing evidence has revealed that topical dexpanthenol can be employed as an effective and well-tolerated agent for AD flare and maintenance [19]. Based on the published evidence and Korean dermatologists’ expert consensus, we aimed to review the experiences, and recommendations on the management of AD, especially in terms of dexpanthenol use.

## 2. Overview of AD and Treatment Options

### 2.1. Prevalence of AD

AD is highly prevalent in children. It affects 15–20% of children (ISAAC study) and up to 3% of adults worldwide [20]. Regions with a high prevalence of AD are characterized by urbanization and industrialization [20,21,22,23]. A recent study has reported that the prevalence of AD in infants aged 1–12 months was 30.48% in China [24]. The prevalence rate based on the Korea National Health and Nutrition Examination Survey (2010–2012) was highest at 3.5% for men and 4.3% for women aged 19–29 years and declined sharply in people aged 30 and above [25]. In Japan, the prevalence of childhood AD was 12–13% in the mainland [22]. As AD has an overwhelmingly high prevalence in children globally, it is necessary to provide effective, safe, and well-tolerated agents that are convenient for daily use.

### 2.2. Pathophysiology of AD

The development of AD is a multifactorial process involving immunologic defects, dysfunctional skin barrier, genetic variations, and environmental factors [2,26,27]. A biphasic inflammation pattern is frequently observed in the disease course of AD. Acute flares are triggered by a Th2-biased immune response, while Th1/Th22 deviation is predominantly responsible for chronic lesions [28]. Stratum corneum (SC), supported by a lamellar-structured extracellular lipid matrix consisting of ceramides, cholesterol, and free fatty acids, plays an indispensable role in preventing transcutaneous water loss and bacterial invasion. Defective skin barrier function, leading to increased transepidermal water loss (TEWL) and decreased SC hydration (dry skin), is a characteristic feature of AD. An impaired skin barrier plays a significant role in various skin conditions, such as dry skin (as a condition itself), sensitive skin, seborrheic dermatitis, contact dermatitis, or AD [29,30].

### 2.3. Burdens of AD

The burden of AD arises from not only the symptoms, but also the chronic course of the disease. AD can involve physical, social, and mental impacts, ultimately worsening a patient’s quality of life. Many patients with AD suffer from itching and pain, leading to significant sleep disturbance, anxiety, and depression [31]. One of the most challenging parts of AD treatment is that it is hard to prevent flares completely and overcome the disease entirely. Recent evidence indicates that AD is not only a dermatological disease but also an inflammatory disease that extends beyond the skin. Patients with AD have a greater risk of cardiovascular disorders than healthy controls, including stroke, myocardial infarction, angina, and peripheral vascular disease [32].

Furthermore, out-of-pocket health care expenses associated with AD are a significant burden on patients with AD [33]. The cost of sophisticated emollient therapies often makes patients hesitant to use the recommended amount of 250 g/week for adults [34].

### 2.4. Treatment Options

Based on recent guidelines and consensus, topical emollient/moisturizer, TCS, topical calcineurin inhibitors (TCI), topical phosphodiesterase 4 inhibitors, oral immunosuppressants, and biologics are current effective treatment options for AD [34,35,36,37,38]. TCS has long been the first criterion of choice for AD therapeutics, generally in combination with topical moisturizers. Despite advances in the development of systemic drugs such as dupilumab, topical therapies continue to be essential for skin barrier dysfunction and for the delivery of anti-inflammatory therapeutics [39].

The daily use of topical moisturizers may help manage AD or can decrease the frequency of flare recurrence. Topical moisturizers might have a role as skincare products during post-inflammatory maintenance stages due to their established skin hydration, skin barrier restoration potential, and wound healing effects [19,40,41]. Guidelines in Asia, the USA, and Europe recommend the daily application of moisturizers as first-line therapy [35,36,39]. Moisturizers should be selected depending on skin type, degree of dryness, and the humidity of the climate [42].

For mild-to-moderate AD, moisturizers, TCS, and antihistamines are generally recommended. For severe AD, more potent TCS, TCIs, systemic immunosuppressants, biologics, and phototherapy are considered. The treatment goal is to achieve absent or mild symptoms without medication, and to reduce or eliminate discomfort in performing daily activities, and slight symptoms can be controlled by moisturizers [43]. In any case, moisturizers and patient education are necessary for the management of AD.

### 2.5. TCS and TCI Treatment for AD

TCS is recognized as a mainstay for AD treatment for mild-to-severe symptoms. TCS produces anti-inflammatory, antipruritic, and vasoconstriction effects via interaction with steroid receptors. The inflammatory cascades during the flares of AD symptoms are suppressed, along with the inhibition of the release of inflammatory mediators. Recent guidelines on AD management from the American Academy of Dermatology, the Joint Task Force, European Task Force on Atopic Dermatitis (ETFAD), and Asia, including Korea, Japan, and China, have recognized TCS and emollients as initial therapy options for targeting inflammation [23,36,39,43,44,45]. For preventive purposes, intermittent proactive application with TCS could help treat frequent flares [39]. However, topical steroids sometimes cause side effects; e.g., a human and murine study has revealed that the short-term (three days) use of TCS (clobetasol 0.05%) could disrupt the epidermal barrier by inhibiting the epidermal synthesis of fatty acids and impairing SC cohesion and integrity, delaying the recovery of the epidermal barrier [46].

TCIs, for example, tacrolimus ointment and pimecrolimus cream, are non-steroid anti-inflammatory drugs. TCIs have been proven effective in short-term, long-term, and proactive treatment of AD. In contrast to TCS, TCIs are not associated with skin atrophy, glaucoma, or cataract, which favors their use in delicate and sensitive areas and for long-term management. TCIs are well-tolerated, but some patients experience a burning sensation and transient worsening of skin conditions, particularly during acute flares [39].

## 3. How Dexpanthenol Can Help in AD

### 3.1. Property and Mechanism of Action of Dexpanthenol

Dexpanthenol is the dextrorotatory isomer of panthenol, and only the dextro-form is biologically active. Panthenol (provitamin B5) and pantothenic acid (vitamin B5) have a similar structure, and the oxidation of panthenol produces pantothenic acid. All animals need pantothenic acid to synthesize coenzyme A (CoA), which plays a crucial role in the oxidation and synthesis of fatty acids [40]. Dexpanthenol is an odorless, transparent, colorless, and highly viscous liquid at room temperature. It is freely soluble in water and alcohol. Its physical properties make it easy to formulate pharmaceutical dosage forms, such as ointments, gels, creams, and hydrogels. Dexpanthenol, pantothenic acid, and their derivatives, are regarded as safe by Cosmetic Ingredient Review [47], and dexpanthenol has been approved by the Food and Drug Administration and the European Commission on Cosmetics.

Since dexpanthenol is highly hygroscopic, it can penetrate easily into the skin and serve as a moisturizer or humectant to maintain the normal skin barrier properties, smoothness and skin elasticity [19,40]. Several in vivo and in vitro studies have shown that dexpanthenol promotes fibroblast proliferation, accelerates re-epithelization, moisturizes the skin, restores the skin barrier, and heals wounds [40,48,49]. In animal studies, dexpanthenol demonstrated cell proliferation and epithelium protection [50,51,52]. Owing to these effects, the combination of dexpanthenol with nasal decongestants could relieve symptoms in patients with acute rhinitis [53]. Dexpanthenol protected against lipopolysaccharide-induced acute lung injury in mice [54].

### 3.2. Effect of Dexpanthenol on Skin Barrier Function

The skin barrier serves as frontline protection, so its intact function and restoration are implicated in various skin conditions including dry skin, sensitive skin, seborrheic dermatitis, AD, and contact dermatitis. Preventive skin hygiene, such as stabilizing skin barrier function with topical treatment, is critical in the care of patients with AD [55]. In most guidelines and consensus, a daily, frequent, regular application of moisturizers, which can help to enhance the skin barrier function, is recommended or required [23,36,39,43,56,57]. Asian countries also consider moisturizers as an important skincare method to provide better skin barrier function [23,36,43,57,58]. Emollients may be composed of humectants for promoting SC hydration and occlusives for reducing moisture evaporation. Although emollients are the basic therapy for skin barrier dysfunction, the direct sole use of emollients on inflamed skin areas is poorly tolerated, and treating the acute flare first is recommended [39].

Due to its extremely hygroscopic characteristics, dexpanthenol provides notable humectant effects. Topical dexpanthenol improves skin hydration and reduces transepidermal water loss (TEWL), thus maintaining the skin’s smoothness and elasticity [40,41,59]. According to the evaluation of average moisture retention for 5 h, dexpanthenol mediates sustained tissue moisturizing effects [60].

Topical 2.5% dexpanthenol formulated in lipophilic vehicles was applied to the skin of 60 healthy volunteers in a double-blind, randomized controlled trial. Dexpanthenol application twice a day for seven days significantly improved SC hydration and reduced TEWL, compared with vehicle controls [41]. In another clinical study, the effect of dexpanthenol cream on skin barrier repair significantly increased after sodium lauryl sulfate (SLS)-induced irritation. After application of dexpanthenol cream for seven days, the skin barrier function was restored. Significant differences were observed between dexpanthenol use and placebo treatment [61]. In addition, SC hydration at dexpanthenol-treated sites remained steady following seven-day treatment with SLS [62]. The hydrating effect may be interrelated with its capacity to regenerate the epidermal barrier [63]. Repairing the skin barrier or preventing barrier dysfunction are essential strategies for reducing the risks for eczema [64].

### 3.3. Effect of Dexpanthenol on AD Flares

As first-line therapy for acute flares, emollients [65] and TCS [58] are recommended for treatment and remission of AD. For acutely inflamed flare lesions, the guidelines indicate that treatment with anti-inflammatory topicals, such as TCS or TCIs, is required first, rather than emollients alone [35,39]. For acute flares, especially oozing and erosive lesions, the ‘wet-wrap’ treatment has been recommended by the ETFAD [39]. For patients with moderate to severe AD, wet-wrap therapy containing emollients with or without TCS could be recommended to relieve pruritis, severity, and improve hydration during flares [35,66]. Wet-wrap therapy is highly effective and could improve tolerance, especially for patients with acute, oozing, and erosive lesions, and for children [44].

Emollients may cause irritation when directly used on inflamed skin. Acute flares should always be treated first with appropriate TCS, followed by emollients and emollients on the surrounding skin [39]. Two conditions are required to ensure the effectiveness of emollients—control of acute flares and proper formulation of the emollient [67]. The order of application of TCS and emollients did not result in significant variation of treatment outcomes. Therefore, emollients can be applied before or after TCS [68]. Application of topical agents is recommended within a few minutes following showering or bathing while a small amount of moisture remains. The consensus is that the topical agent should be left for an appropriate period to allow complete absorption before applying another agent. The guidelines recommend the “soak and smear” technique when using topical agents (emollients and/or TCS) to maximize the absorption of active ingredients, which penetrate the epidermal layer via the expanded pores resulting from bathing [29,35,69,70].

Applying emollients as freely and frequently as possible is recommended, preferably every 4 h or at least 3–4 times per day. The ETFAD recommendations indicate that a sufficient quantity of emollients, at least 30 g/day or 1 kg/month for an adult with AD, should be applied in a ‘soak and smear’ or ‘soak and seal’ technique [44].

### 3.4. TCS Sparing Effect of Dexpanthenol

Previous results have suggested that the effectiveness of dexpanthenol is comparable to that of TCS. The topical application of moisturizers in adequate amounts, irregularly or continuously, was proved effective in sparing the use of TCS as short- or long-term treatment, and in maintaining the remission obtained with corticosteroids [36,71,72,73]. In practical guidance from a national expert panel in Italy, the TCS sparing effect was confirmed when patients were administered moisturizers and emollients intermittently or continuously in appropriate amounts [71]. The supply of hydration was also proved by the sustained remission of atopic lesions obtained with TCS treatment. Optimal skin hydration could reduce skin inflammation and the frequency of flares [71]. Eventually, the amount of TCS was decreased following the topical application of moisturizers [36]. The application of both agents twice daily is the basic recommendation of most guidelines or consensus [36,39,42,69].

Regular daily use of topical emollients could reduce the amount of TCS for short- and long-term treatment in mild-to-moderate AD [56]. Since the topical application of 5% dexpanthenol showed comparable efficacy with low potency TCS, dexpanthenol could be a substitute for TCS [73]. According to the evidence of steroid-sparing, the expert panel recommended that when dexpanthenol is used on a daily basis, TCS can be used every other day, particularly for infants, children, or patients with potential TCS side effects or steroid-phobia.

For severe AD, TCS or TCI is required to achieve effective treatment. Proper guidance, persuasive education, and basic treatment such as the use of dexpanthenol would be required for patients who abuse TCS, including addiction to high potency TCS, or have steroid-phobia [74].

Overall, the panel recommended that patients with atopic dermatitis should use TCS and dexpanthenol alternatively, especially if the disease is mild to moderate.

### 3.5. Special Populations

Schmutz et al. reported non-inferiority in maintaining TEWL scores in acute radiation dermatitis (0.1% methylprednisolone cream vs. 0.5% dexpanthenol cream) [75]. As shown in full-thickness 3D skin models representing acute radiodermatitis and mucositis, skin impairment seven days after radiotherapy demonstrated a completely restored epidermal part after treatment with dexpanthenol-containing ointment or liquid [76].

Gestational AD is one of the most common skin diseases during pregnancy. According to Japanese guidelines, TCS is considered safe for both pregnancy and breastfeeding, since the absorption of TCS into the bloodstream is low [43]. In the Taiwanese consensus, the application of TCS during pregnancy was also disclosed as safe, except for fluticasone propionate due to its metabolic characterization [57]. Although normal use of lower potency TCS is regarded as safe, low birth weight might be related to long-term use of higher potency TCS at high doses (≥300 g) [43]. Dexpanthenol has therapeutic effects on nipple trauma through epithelialization and granulation [77]. During the lactation period, TCS should be smeared after breastfeeding, followed by the cleaning of nipples before feeding [57]. The use of emollients and TCS with a moderate-to-low potency is the ideal treatment for this area [78]. Skin damage can be caused by ablative laser therapy, microneedling, or tattooing. To facilitate wound healing, dexpanthenol reduces inflammation, and promotes cell proliferation and epithelial remodeling [49]. Clinically, dexpanthenol demonstrated superior re-epithelialization rates compared to standard treatments such as petroleum jelly. Therefore, topical dexpanthenol is recommended as an effective treatment option for superficial skin damage in the early stages [49]. Among patients who received laser corneal surface ablation, 2% dexpanthenol resulted in significantly better vision and reduced residual cylinder after seven days, compared to artificial teardrops [79]. Topical application of 5% dexpanthenol to freshly tattooed skin restored the skin barrier, as demonstrated by TEWL [80]. Throat pain and tonsillar wound healing after tonsillectomy were significantly improved in dexpanthenol-treated patients, via its anti-inflammation, skin hydration, and mucosal protection properties [81]. Dexpanthenol could prevent the occurrence of postoperative sore throat [82], and promoted skin healing at the laser-irradiated site of photo-damaged skin [83]. Dexpanthenol produced significantly improved results and re-epithelialization earlier after laser therapy. Hence, dexpanthenol could be a promising alternative to routinely used treatments for wound healing.

Furthermore, cheilitis associated with isotretinoin treatment was markedly improved after topical application of 5% dexpanthenol cream [84]. The expert panel recommended dexpanthenol ointment as an effective and well-tolerated treatment for cheilitis, due to its hygroscopic activity and moisturizing formulation. Another study reported that 0.5% dexpanthenol cream in addition to TCS delayed the development of acute radiation dermatitis, in terms of clinical scores and TEWL values [75].

### 3.6. Effect of Dexpanthenol on AD Maintenance

AD depicts various phenotypes and endotypes depending on age, chronicity, atopic status, or ethnicity [28,85]. For long-term maintenance therapy to maintain adequate skin hydration and prevent flares, moisturizers such as dexpanthenol ointment should be continuously used, at least twice daily, after the induction of remission by TCS [39,58].

In the recent study involving infants and children with stabilized mild AD [86], a dexpanthenol medical device cream was applied 2–3 times daily in the stabilization phase, until severity was reduced. Then, a topical panthenol-containing cosmetic emollient was used twice daily during the maintenance phase. At the end of the three-month study, the proportions of patients without flares in the dexpanthenol and reference groups were 96% and 77%, respectively. In healthy subjects, the same dexpanthenol-containing emollient was effective in reducing TEWL and enhancing skin hydration. Moreover, Raman spectroscopy revealed that dexpanthenol-containing emollient was associated with sustained and deep skin moisturization and improved intercellular lipid lamellae organization [48,87]. Higher water distribution was observed by the relocation of the water molecules from more superficial to deeper layers of the SC, which resulted in deeper moisturization [87]. Dexpanthenol formulation effectively increased skin hydration and was well-tolerated in healthy infants between 3 and 25 months old without significant change in mean cutaneous tolerability scores [87].

### 3.7. Emollient Plus in Maintenance

Emollients are mostly recommended as the first-line therapy for AD, even when the disease is clear or almost clear, as well as during acute flares and remission [39,43,65,70,88]. Proksch et al. proposed that emollients should include a proper combination of humectants, physiological and non-physiological lipids, antipruritics, and multifunctional components such as dexpanthenol [89]. Selecting an appropriate emollient for patients with AD would improve acceptability and adherence for emollient treatment. A physician’s recommendation is the primary consideration for patients when choosing an emollient; therefore, doctors should provide evidence-based information on these emollients [90].

Traditionally, emollients are generally considered to be topical formulations without active pharmaceutical ingredients. “Emollient plus” has been defined to include topical formulations with vehicle-type substances and additional active, non-medicated substances [39]. The active, non-medicated substances are active ingredients that do not qualify as topical drugs [66], which include saponins, flavonoids, vitamins such as riboflavin and niacin, and beneficial bacterial lysates. Some of these active ingredients were found to improve skin protection, relieve pruritis, exert an anti-inflammatory reaction, exhibit antioxidant properties, and provide biologically essential lipids and antimicrobial activities [91,92,93]. Dexpanthenol is approved as a cosmetic ingredient with established safety, and it has evidence-based potent skin-hydration and wound-healing effects [19,40,41]. Products containing dexpanthenol as active ingredients can be considered “emollient plus”.

Dexpanthenol accelerated the wound healing process (by a factor of 1.52 vs. the vehicle) and promoted fibroblast proliferation, in vivo and in vitro [40]. Another double-blind study monitored by histological examination revealed that dexpanthenol accelerated the wound-healing process [94]. In a randomized controlled trial, wound healing effects of water-filtered infrared-A (IRA) and/or dexpanthenol were examined in 12 healthy subjects using an acute wound model. Measured by laser scanning microscopy, the fastest SC formation was observed when water-filtered IRA irradiation was combined with dexpanthenol cream [95].

For healthy volunteers with dry skin, topical dexpanthenol-containing emollients (oil-in-water formulation) were topically applied like cosmetic products for daily care over four weeks [59]. The dexpanthenol formulation induced a significant increase in skin elasticity as measured by Cutometer^®^ MPA580 (Courage & Khazaka, Cologne, Germany), skin hydration, TEWL, and SC lipid contents. Use of the dexpanthenol formulation once daily for over 28 days was well-tolerated in healthy adults.

Other formulations with dexpanthenol were tested in healthy adult women who underwent non-ablative laser resurfacing, laser depilation, or chemical peel [96]. The tested formulations maintained skin integrity, promoted recovery of damaged skin, and reduced erythema, and were associated with significantly decreased TEWL and dermal temperature. In the field of aesthetic dermatology, these dexpanthenol-containing formulations were well appreciated and would be an appropriate option for post-procedural care.

Compared with drug-free vehicles, panthenol-containing emollient plus can provide additional benefits such as accelerated wound healing, more prominent skin hydration, reduced skin redness from inflammation, and improvement to rough skin [62]. 5% dexpanthenol cream was superior to placebo in terms of SC hydration and protection against skin irritation in 23 healthy participants after exposure to SLS.

## 4. Conclusions

The treatment of AD requires long-term, risk-based stepwise management. Moisturizers are the first-line or basic therapy for AD treatment across various national guidelines and consensuses. Topical application of dexpanthenol significantly improved SC hydration and skin barrier function compared with the control. Appropriate use of dexpanthenol ointments during acute dermatitis flares is useful for minimizing epidermal disruption caused by TCS. The regular use of a dexpanthenol ointment subsequent to remission of AD flares has a steroid-sparing effect. The current evidence reveals that 5% dexpanthenol ointment has a good efficacy, safety, and tolerability profile, and is suitable for use during pregnancy and lactation.

Panthenols rapidly convert to pantothenic acid, resulting in very low toxicity. Allergic or irritant reactions to dexpanthenol have been reported, but overall it is generally well-tolerated [40,97]. As with all medications, proper use of dexpanthenol should be discussed with physicians.

Dexpanthenol-containing emollients, especially water-in-oil formulations, are considered “emollient plus” for AD treatment and provide improved skin hydration and wound healing effects compared with conventional emollients. Summary of the current evidence indicates that dexpanthenol might be a suitable ingredient for flare control and maintenance of AD. Physicians should consider prescribing an emollient plus over TCS if patients have steroid-phobia or show signs of TCS side effects. Because atopic dermatitis needs long-term management, dexpanthenol could be a promising ingredient for patients.

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
