# Peer review of "Use of Dexpanthenol for Atopic Dermatitis—Benefits and Recommendations Based on Current Evidence"

_jcm, 2022, doi:10.3390/jcm11143943_

Round 1

Reviewer 1 Report

Thank you for your interesting of our journal. Please revise belows.

Itroduction

-> Focus onemollent action in AD and please shortened the length. 

Overview of AD and treatment options

-> Please add "Cardiovascular comorbidities of atopic dermatitis: using National Health Insurance data in Korea" to burdren of AD.

-> Please add TCI treatment of AD

How dexpanthenol can help in AD 

->  Please shortened the length. 

-> "Effect of dexpanthenol on AD flares, TCS sparing effect of dexpanthenol, Special populations, Effect of dexpanthenol on AD maintenance" recommend to write  "Errect on dexpanthenol on AD treatment" by reducing the contents at once.

Reviewer 2 Report

The manuscript „Use of dexpanthenol for atopic dermatitis…“ gives an update on past and current literature on the use of the well established compound dexpanthenol.

The beneficial effects of dexpanthenol are summed up convinvcingly - as to barrier improvement (anti-drying as well as protection againt intruding germs), acceleration of wound-healing and reepithelialization and more. Several observational studies are reported, covering past 20 years.

The manuscript is well structured and written in very high quality - like a textbook chapter: close to perfection.

The more general, introductory part on AD and its treatment options should be shortened for readers from dermatology field.

It is hard to believe, that such an effective substance does not have any side effect. Please name at least one disadvantage.

Author Response

The more general, introductory part on AD and its treatment options should be shortened for readers from dermatology field.

Our Response: Thank you very much for your valuable comment. We have shortened the aforementioned parts to be more succinct.

It is hard to believe, that such an effective substance does not have any side effect. Please name at least one disadvantage.

Our Response: We thank the reviewer for raising an important point. As dexpanthenol is the isomer of panthenol (provitamin B5) and panthenols rapidly convert to pantothenic acid resulting in very low toxicity, it is associated with few adverse events. However, allergic or irritant reactions to dexpanthenol have been reported. We have added the related information on side effects in the revised manuscript accordingly.

In the revised manuscript:

Panthenols rapidly convert to pantothenic acid resulting in very low toxicity. Allergic or irritant reactions to dexpanthenol have been reported, but overall it is generally well-tolerated (41, 98). As with all medications, proper use of dexpanthenol should be consulted by physicians.

We sincerely thank the reviewer for the time and efforts on this manuscript.

Reviewer 3 Report

The title describes atopic dermatitis, in the text various other diseases occur

The following terms should be explained, they are mentioned again and again in the text and the reader wonders where the differences are:

- emollients

-  moisturizer

- humectants

Author Response

The following terms should be explained, they are mentioned again and again in the text and the reader wonders where the differences are:

- emollients

-  moisturizer

- humectants

Our Response: We thank the reviewer for raising an important point. The terms emollient and moisturizer are interchangeable. They refer to a substance/product that moisturizes and smooths the skin.  On the other hand, humectant is an ingredient that increases or maintains hydration of the skin. We have supplemented the related information in the revised manuscript accordingly.

In the revised manuscript:

Moisturizer is used synonymously with emollient and refers to as a product that moisturizes and smooths the skin, whereas humectant increases or maintains hydration of the skin (18).

We sincerely thank the reviewer for the time and efforts on this manuscript.
